

**Real time rainfall estimation using microwave signals of cellular communication networks:**
**a case study of Faisalabad, Pakistan**
Muhammad Sohail Afzal[1], Syed Hamid Hussain Shah[1], Muhammad Jehanzeb Masud Cheema [1,], Riaz Ahmad[2]
[1]Department of Irrigation & Drainage, Faculty of Agricultural Engineering& Technology, University of Agriculture Faisalabad, Pakistan
[2]Department of Agronomy, Faculty of Agriculture, University of Agriculture, Faisalabad, Pakistan
Correspondence to: Muhammad Sohail Afzal (msohail1227@yahoo.com)
**Abstract**
Water balance estimate requires high spatio-temporal water balance components and rainfall is
one of them. Rainfall is stochastic variable, which varies with respect to space and time. There
are  different methods for rainfall estimation such as rain gauge, satellite data but the resolution
of these methods are very low, which cause over and underestimation of rainfall. A real time
rainfall estimation mechanism is tested using commercial cellular networks in Faisalabad, district
of Pakistan. The microwave links are used to quantify rainfall intensities and estimate rainfall at
high spatio-temporal resolution. The attenuation in electromagnetic signals due to varying
rainfall intensities is measured by taking difference between the power transmitted and power
received during rainy period and is the measure of the path-averaged rainfall intensity. This
rainfall related distortion is converted into rainfall intensity. This technique is applied on a
standard microwave communication network used by a cellular communication system,
comprising 35 microwave links, and it allow for observation of near-surface rainfall at the
temporal resolutions of 15 min. Signal data-set of year 2012-2014 and 2015-2017 is used for
calibration and validation respectively with three rain gauge data-set. The accuracy of the
method is demonstrated by comparing the  daily cumulative rainfall depth of University of
Agriculture Faisalabad rain gauge (UAF-RG), Ayub Agriculture Research rain gauge(AR-RG)
and Water and Sanitation Authority rain gauge  (WASA-RG) with link based rainfall depths
estimated from L2, L28 and L34 respectively, reaching $r^2$ up to 0.97. UAF-RG is considered
reference to study the spatial variability of rainfall of all the selected links within the study area,
observed 10%-60% average spatial error of all links with the reference UAF-RG. All the results
show that microwave links are potentially useful compared to the low resolution methods of
rainfall estimation and can be used for effective water resources management.
**Keywords**: Rainfall, microwave signal data, Interpolation, rain gauge data and path average
rainfall intensity



| Nomenclature | |
|---|---|
| PL | Path length (distance between two towers) |
| D-(UAF-RG) | Distance of link from University of Agriculture Rain Gauge |
| CV | Coefficient of variance |
| $r^2$ | Coefficient of determination |
| IDW | Inverse Distance Weighted |
| TRMM | Tropical Rainfall Measuring Mission |
| GPM | Global Precipitation Measurement |
| PMD | Pakistan Meteorological Department |
| D-(AR-RG) | Distance of link from Ayub Research Rain Gauge |
| D-(WASA-RG) | Distance from Water and Sanitation Authority Rain Gauge |


## 1. Introduction

The management of water resources requires high temporal and spatial information of
rainfall.   Rainfall is considered as an important input parameter for hydrological model that's
why it needs to be managed and measured very carefully on high spatial and temporal basis. Any
small error of a large water balance component that is rainfall can produce significant error in the
small components such as runoff, leaching, capillary up flow from shallow groundwater.
Without exact measurement of rainfall, agricultural crops, surface and groundwater resources
cannot be managed on sustainable basis (Yilmaz el al., 2005; Berndtsson and Niemczynowicz

41 1988).

Aerial rainfall for catchment and basin is normally interpolated from rain gauge, radar and
satellite data but these sources provide very low resolution data and all these instruments has
their own challenges. The spatial interpolation of point measurements in heterogeneous
landscapes and mountains result in erroneous estimates. Dense networks are needed that are
difficult to establish and maintain in under developing countries. Rainfall estimation by using
satellite data cannot provide full coverage of rainfall due to low spatio-temporal resolution.
There are geostationary satellite observations are available having temporal resolution of 15 min
but are often very indirect e.g. estimates through cloud physical properties (Roebeling and
Holleman 2009). There is another new product of NASA called GPM mission having spatial



resolution 0.1$^o$ and temporal resolution of 30 min but this is still very low resolution as compared
to the rainfall estimated by microwaves links (Hou et al., 2014; Rios Gaona et al., 2016).
Similarly another data source which is mostly used is TRMM having spatial resolution 0.25$^o$ and
temporal resolution minimum 3 hours .The downscaling of satellite image is a technique that can
estimate rainfall for a smaller range of distance, but this technique is indirect technique and
produces biasness and uncertainty in results. The rainfall estimated from radar normally
deteriorates for longer ranges from radar.

58       In Pakistan rain gauge networks are managed and operated by Pakistan Meteorological

Department (PMD). There are total 97 rain gauge stations in Pakistan which include 28 in
Punjab, 19 in KPK, 05 in Azad Kashmir, 09 in Northern Areas, 27 in Sindh-Balochistan and 09
observatories controlled by Geophysics Quetta which are insufficient to capture high spatio-
temporal rainfall (PMD Website). The intensity of telecommunication tower is greater than the
magnitude of rain gauges. Therefore the main focus of the study is to quantify the rainfall due to
the signal attenuation and promote the use of microwave links for rainfall estimation in Pakistan.

65       The basic concept behind the rainfall estimation from signal attenuation is that the signal

that travels between the two towers attenuates due to rainfall intensity and this attenuation
depends upon the rainfall duration and rainfall intensity, as more intensity of rainfall will create
more signal distortion. As the number of raindrops and intensity of rainfall increases, the
attenuation of link also increases, which subsequently reduced the received power at the other
end of the link. The power received at the other end of the link is considering a byproduct of the
communication between the networks (Zinevich et al., 2008; Goldshtein et al., 2009; Zinevich et
al., 2009; Overeem et al.,(2011,2013,2016);Messer et al., 2006; Leijnse et al., 2007b; Zinevich et
al., 2009; van het Schip et al. 2017, Rios Gaona et al., 2015). This new advancement of using
link data for rainfall estimation is very helpful to estimate the rainfall at a very high resolution
which will further use for flood prediction, drought management, crop productivity and risky
climate warning. Similarly massive deployment of these microwave links provides a
complementary network to measure rainfall, especially in countries where rain gauges are scarce
or poorly maintained, and where ground-based weather radars are not (yet) deployed
(Doumounia et al., 2014).

80       In section 2, the detail of cellular communication link and rain gauge data information is

explained. In section 3, description about how to use signal data to measure path-averaged

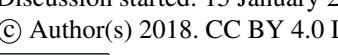



rainfall intensity, mapping technique for rainfall map and other methodology to study the spatial
variability of rainfall and in section 4, final discussion and conclusion is elaborated.
**2. Type of data used**
**2.1. Microwave link data**
In order to estimate the path-averaged rainfall intensities, signal data of 35 selected links in
Faisalabad is obtained from international telecommunication Network Company Telenor,
working in Pakistan. The maximum and minimum received power over 15 min temporal
resolution having 38 GHz frequency is used. Figure 1 explains the location of 35 selected
microwave links and location of rain gauges. It is clear from the Fig.1 that L2, L28 and L34 are
close to UAR-RG, AR-RG and WASA-RG respectively. All the selected links are vertically
polarized and in the radius of 225 km$^2$ area. The data format required to process the code is
acquired from Overeem et al. (2016). Total 32 and 33 days from years 2012-2014 and 2015-2017
are selected for calibration and validation of link based approach with standard rain gauges data-
set respectively. The path length, i.e. distance between the sender and receiver, for all the
selected links is between 0.50-2.5 km.

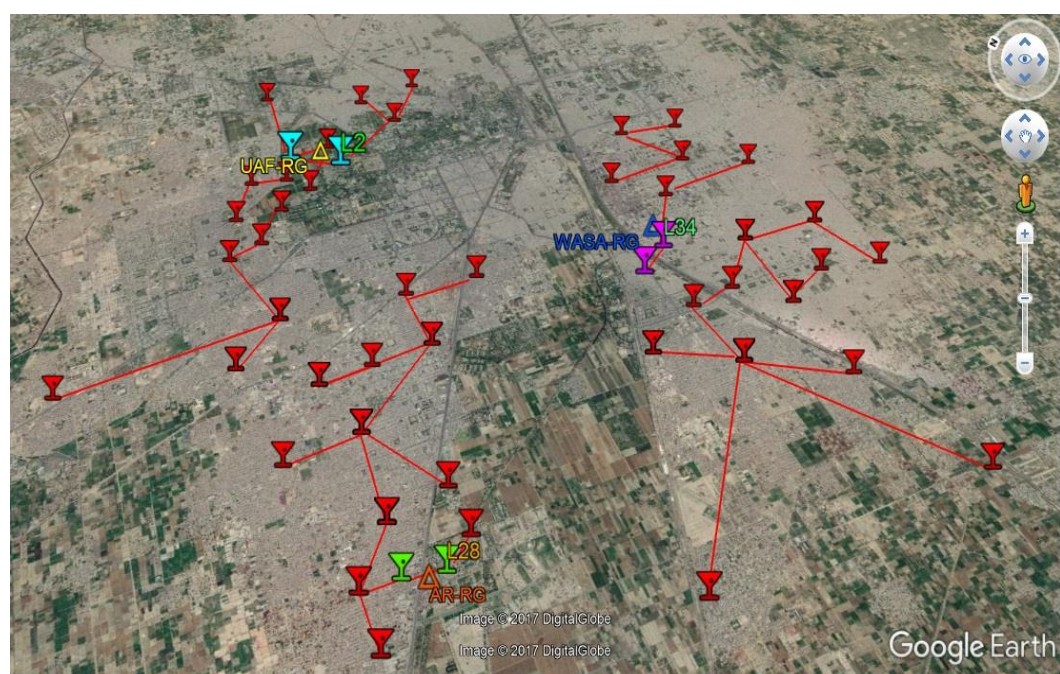




**Figure 1**. Map of study area with location of selected microwave links (red, zinc, green and purple towers), and triangles: yellow color (University of Agriculture), orange color (Ayub Research) and blue color (Water and Sanitation Authority) rain gauge.

## 2.2. Rain gauge data

The employed rain gauges data are obtained from three different rain gauge stations namely UAF-RG, WASA-RG and AR-RG, operated by University of Agriculture Faisalabad, Ayub Agriculture research and Water and Sanitation Authority Faisalabad respectively. The data provided by all these rain gauge stations are daily cumulative rainfall values against each rainy day. For calibration and validation purpose, signal based rainfall having 15 min resolution data is converted into daily cumulative rainfall value for each day to compare it with UAF-RG, WASA-RG and AR-RG stations. Independent calibration and validation of three selected links i.e. L2, L28 and L34 against UAF-RG, AR-RG and WASA-RG respectively are performed and by considering UAF-RG as a reference point, calibration and validation of all the selected links against UAF-RG are performed to study the spatial variability of rainfall

## 3. Methodology

### 3.1. Rainfall retrieval for signal based rainfall

Microwave links are the main source of communication between the telecommunication towers and these microwave links attenuate due to rainfall intensity (Upton et al., 2005). This distortion in the signal can be measured by using power law studied by Atlas and Ulbrich (1977) which is the relationship between rainfall and specific attenuation

$$r = cz^b \tag{1}$$

In Eq. (1) $z$ is the specific attenuation, $r$ is the intensity of rainfall ($mmh^{-1}$), $c$ is the coefficient and $b$ is the exponent and the values of these coefficient and exponent depend upon polarization, frequency of the signal, temp of the surrounding, water phase and other important factor which include drop size distribution, canting angle distribution and shape of the rain drop (Jameson, 1991; Berne and Uijlenhoet, 2007; Leijnse et al., 2010a) and further explained by Overeem et al., (2016). During rainy period the entire length (km) of the signal between the two tower attenuate (dB) and thus the intensity of the rainfall is given by



$$F_{ref}(L) - F(L) = Am = \int_0^L z(d)\,ds = \int_0^L \left[\frac{r(d)}{c}\right]^{1/b} \tag{2}$$
In Eq. (2) $F_{ref}$ is the reference signal level, d stands for the entire length of the signal and F
(L) is the received power (dBm). After approximation final form of power law is given below
(Overeem et al., 2011, 2013, 2016)
$$<r> = c\left[\frac{Fref(L) - F(L)}{d}\right]^b \tag{3}$$
The value of coefficient c and exponent b as explained by Overeem et al., (2016). Berne and
Uijlenhoet (2007) studied that how link length, frequency, precise drop size division effect the
average rainfall intensity for links having frequency range between 12 to 38 GHz. They
concluded that the value of coefficient c and exponent b will depend on the frequency of the link
and not account much on the length of the signal. If the length of the signal increases, the
frequency of the microwave link decreases. The reason behind is that if length increases, than the
effect of rain drop on the frequency does not gives the best result, therefore links are usually
selected within acceptable distances for making sure strong signal strength.
The concept of rainfall estimation is derived from the minimum and maximum received
power having 15 min high resolution. This maximum and minimum received power is converted
into corrected minimum and corrected maximum received power by comparing with reference
signal power. In the first step, the pre-processing of link data is done using the code developed in
R software (Overeem et al., 2016). In this step, the signal data of previous day and present day is
converted into one file based on the selection of links having frequency of 12-42 GHz. If a
unique link contains more than one record, that link is removed during the pre-processing,
because one unique link can have only one record for a specific time interval. Also in this step, it
is confirmed that whether frequency, link coordinates, and path length of a unique link remains
same in whole day. This criteria is very important because these parameters should not change
during a day, if this is the case, that link is also removed.
Based on above checks, one file is prepared which is free from errors. In the next step, the
file prepared in the first step is used for further processing related with categorization of wet and
dry signals using the code in R developed separately for this step (Overeem et al., 2016). In the
next step, the link having both ends within 15 km from either side end selected link. Based on the



threshold value of signal, the wet and dry signals are identified (Overeem et al., 2011, 2016). In
the third step rainfall intensities are estimation based on the corrected maximum and minimum
received signal power of the file prepared in the above step using the power law relationship
(Overeem et al., 2011,2016), Leijnseet al., 2007, van het Schip et al. 2017, Rios Gaona et al.,

158  2015).

There are many type of errors that may come in the way to estimate the rainfall intensity and
these errors may be because of the reflection and refraction of the beam, dew formation on the
surface of the antennas, antenna icing, scintillation, multipath, reliable absorption by the
atmosphere constituents (Upton et al., 2005). According to Upton et al. [2005] there is a very
small fluctuation in the received signal power during dry season as compared to the fluctuation
in the received power when there is no rain. There is another source of error in rainfall estimation
because of the water films on the tower antenna .This  type of  signal attenuation  is  a major
source of  error which is  modified by ( Kharadly and Ross 2001; Minda and Nakamura 2005;
Leijnse et al., 2007a, 2007b, 2008). When there is large distance between the link the change due
to wet antenna is very small because the signal attenuation due to rainfall is very small (Leijnse
et al., 2008).
The temporal sampling describes the number of sample per unit time and used this for
collection of the samples. Leijnse et al. (2008) explained the three type of sampling strategies,
which is averaged, intermittent and continuous. The intermittent and averaged strategies are
mostly used for cellular communication link monitoring. In these two types of sampling
strategies, signal power is observed over averaged 15 min resolution or sample may be selected
in the middle of 15 min period. The intermittent sampling strategies has been used in the
research for rainfall estimation, which is similar to the Messer et al. (2006) which is the
maximum and minimum received power, $F_{min}$ and $F_{max}$ are collected over 15 min resolution.
There is another error that may occur and is responsible for the decrease the availability of data
is due to the heavy rainfall. This type of error may be due to the storage issue arrives in the
server of the telecommunication company.  Overeem et al. (2016) suggested some fixed
parameters on the basis of these errors and all these recommend parameters values are used in
the paper.




### 3.2. Verification methodology

For calibration and validation purpose path-averaged rainfall intensities estimated from the signal data having 15 min resolution are converted in to daily (24hrs) cumulative rainfall value against each day to compare it with daily (24hrs) cumulative rainfall values of UAF-RG, AR-RG and WASA-RG. Independent calibration and validation of L2, L24 and L34 are performed against UAF-RG, AR-RG and WASA-RG respectively.

### 3.3. Percentage error analysis

By considering UAR-RG as reference point to study the spatial variability of rainfall in the study area, calibration and validation is performed for all 35 no of selected links between UAF-RG and signal based rainfall depths. After estimated signal based rainfall, percentage error for all the all selected links is calculated according to the Eq. (4) and Eq. (5), where d is cumulative signal based rainfall, f is cumulative UAF-RG rainfall depth, PD is percentage error of each day and L no is link number.

Percentage error analysis for each selected link against each day (PD) $= \left(1 - \left(\frac{d}{f}\right)\right) * 100$   (4)

Average percentage spatial error for all selected days for each link $= \frac{\sum_{i=1}^{n} PD_i}{L\,no}$   (5)

### 3.4. Rainfall mapping

Path-averaged rainfall estimated from cellular microwave links are spatially interpolated to obtain the rainfall maps. Overeem et al., (2016, 2015, 2013); Rios Gaona et al., (2015) suggested two type of interpolation techniques i.e. ordinary kriging (OK) and inverse distance weighted (IDW). Both these interpolation methods are well suited for dealing with spatially disturbed data locations. The ordinary kriging requires variogram model, so it is not possible to reboust such variogram in this study because of limited data-set. IDW technique is used to interpolate the rainfall maps of study area. The path-averaged rainfall estimated from link approach is considered at the center of the sender and receiver, so that point data can be used in IDW interpolation. Rainfall maps are prepared in GIS by using IDW interpolation technique to study spatial variation in rainfall estimates between signal and rain gauge rainfall depths.



## 4. Result and Discussion

### 4.1. Maximum, minimum received, corrected maximum and minimum received power

The signal attenuation due to rainfall is the main factor to find the rainfall intensities which is the difference between the received signal level and some reference signal level which is representation of the dry period when there is no rain. The attenuation in the signal is estimated by using the procedure explained in the section 3 and compared that attenuation with the reference signal power to find corrected maximum power received and corrected minimum power received (Overeem et al., 2015, 2016). Overeem et al., (2016), van het Schip et al. (2017), Rios Gaona et al., (2015) presented graphs showing the minimum and corrected minimum received power compared with gauge-adjusted radar having 15 min resolution but due to no data availability of radar with same 15 min temporal resolution, it is not possible to make such comparison in this study so only attenuation due to rainfall intensity is shown in fig.2. The Fig. 2 shows the maximum and minimum power received and corrected minimum and maximum power received compared with reference signal level. The top (right, middle and left) plots present attenuation due to rainfall of three different links and bottom (right, middle and right) plots present corrected maximum and minimum received power which is compared with reference signal level. It is clear from Fig 02 that all the links show the different attenuation due to rainfall intensity but the time of distortion remains the same in all the links, which is located index number 66 to 70 time interval.





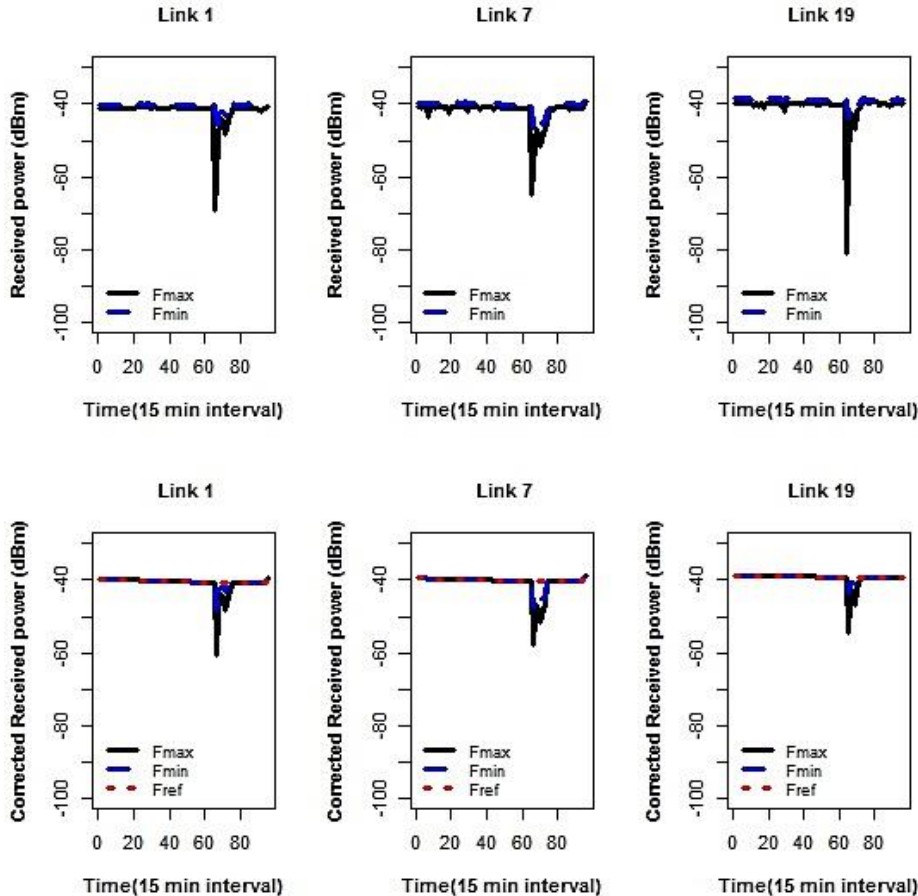

230

**Figure 2**. Top (right, middle and right) plots present maximun recevied power (black line) and minimun recevied power (blue line) for three different links for 12 May 2014. Bottom (right, middle and left) plots present maximun corrected recived power (black line) and minimun correted recevied power (blue line) and reference signal level (red line) for three different links for same day dated 12 May 2014. There are total 96 time intervals having 15 min resolution in each plot against each day (24 hours).

237

238





**4.2. Calibration and validation of signal based rainfall with standard rain gauge data operated in Faisalabad.**

The rainfall estimated from link based approach having 15 min resolution is converted into daily cumulative rainfall depth to compare it with the daily cumulative rainfall depth of rain gauge operated by different institutions in Faisalabad. The comparisons are observed for three different cases of spatiotemporal aggregation, for the daily commutative rainfall depths of UAF-RG, AR-RG and WASA-RG with link based rainfall depths estimated from L2, L28 and L34 respectively. This study tests the performance of the 32 days for calibration and 33 from validation from years 2012-2014 and 2015-2107 respectively. Figure. 3 Top (right, middle and left) plots and bottom (right, middle and left) plots present calibration and validation of links with standard rain gauges. All the scatter plots in fig.3 summarizes the values of the mean rainfall depth ($R_{Link}$, $R_{UAF-RG}$, $R_{AR-RG}$ and $R_{WASA-RG}$), the coefficient of variation (of the residuals) CV, and the coefficient of determination $r^2$ (i.e. the squared correlation coefficient) for the three cases of spatiotemporal aggregation, for link based and standard rain gauges rainfall depths. It is clear from fig. 3 that coefficient of determination for L2, L28 and L34 compared with UAF-RG, AR-RG and WASA-RG are 0.98, 0.96 and 0.95 respectively for calibration data-set, whereas coefficient of determination for L2, L28 and L34 compared with UAF-RG, AR-RG and WASA-RG are 0.97, 0.98 and 0.97 respectively for validation data-set.





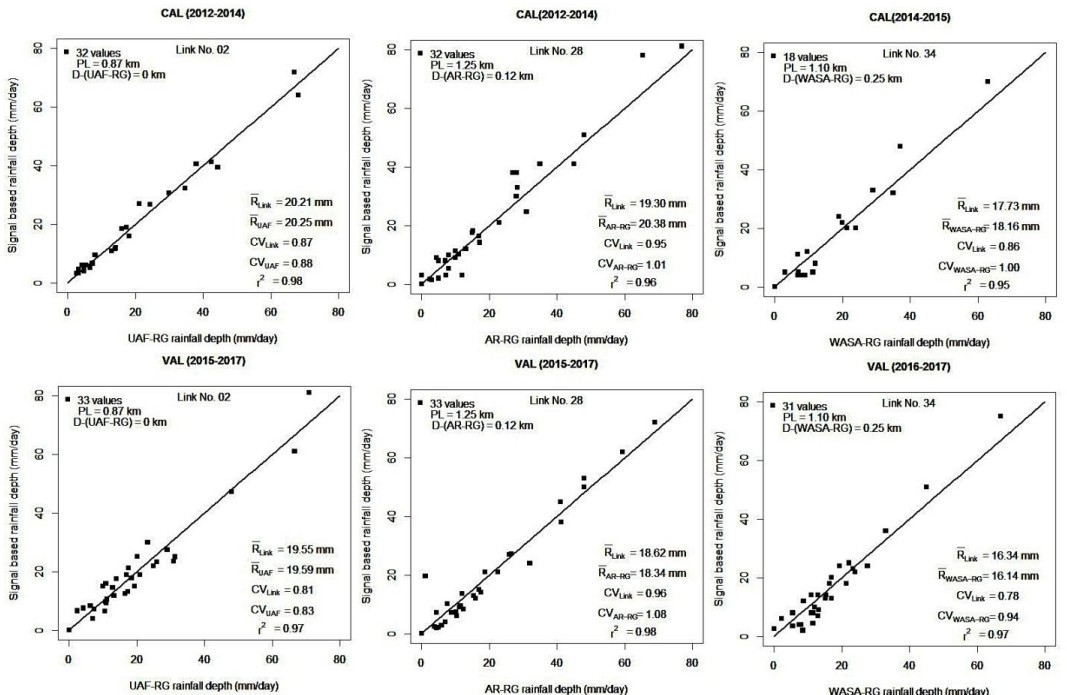

257

**Figure 3**. Calibration and validation of signal based rainfall with standard rain gauges operated in Faisalabad. Left (top and bottom) plots present calibration and validation of L2 with UAR-RG, middle (top and bottom) plots present calibration and validation of L28 with AR-RG and right( top and bottom) plots present calibration and validation of L34 with WASA-RG.

**4.3. Calibration and validation of signal based rainfall for the all selected links with UAF-RG to study the spatial variability of rainfall.**

**4.3.1. Calibration and validation**

The UAF-RG is used as a reference point to study the spatial variability of rainfall in the selected study area. The cumulative rainfall depths of UAF-RG are compared with all the selected 35 links based rainfall depth to study the spatially variability within area of 225 $km^2$. Overeem et al. (2011, 2013, and 2016), Van het Schip et al (2017), Rios Gaona et al., (2015) used a gauge-adjusted radar data set to calibrate and validate the microwave link rainfall retrieval algorithm but in this study due to limited data availability of radar, daily cumulative rainfall values of UAF-RG are compared with the daily cumulative rainfall depth measured by link based



approach for all the selected links. For calibration purpose, total 32 numbers of days are selected
for years 2012-2014. The distance of all the selected links from the reference UAF-RG and
distance between the transmitter and receiver of all the links are measured. The links 02 was very
close to reference UAF-RG nearly 0 km distance and all the other remaining points are in the
area of 225 km$^2$ around the UAF-RG.

277         The comparisons are carried out on the basis of scatter density plots and three metrics:

mean rainfall, coefficient of variation (CV), and coefficient of determination ($r^2$). Figure 04
explains scatter density plots between the daily commutative signal based and daily cumulative
rainfall of UAF-RG station rainfall depth (mm/day). The statistical analysis between observed
UAF-RG and signal based rainfall is analyzed. The values of CV, $r^2$, and the average
commutative rainfall measured using UAF-RG, as indicated by $R_{UAF}$ and average commutative
rainfall depth using signal approach, indicated by $R_{LINK,}$ are included in the plots. The coefficient
of variation CV and coefficient of determination for link which is close to the UAF-RG show
significant results, but as the distance of links increases from the reference UAF-RG, level of
significance decreases. It is clear from the fig.3 that for L7, L15, L18, L22, L23 and L29, as the
distance increases 0.34 km, 3.39 km, 4.13km, 6.21 km, 5.67 km and 10.39 km respectively from
Reference UAF-RG, level of significance i.e. coefficient of determination deceases. For
calibration data-set, the coefficient of determination for L7, L15, L18, L23, L22 and L29, are
0.97, 0.82, 0.79, 0.91, 0.79 and 0.67 respectively, Similarly data-set having frequency 38 GHz is
used in this study for the validation purpose. The same links are used for the validation purpose
as used from calibration purpose but data-set used are of different time period. For validation
purpose data-set of years 2015-2017 are used and total 33 rainy included non rainy days are
selected. Figure 05 explains validation scatter density plots between the daily cumulative signal
based and daily cumulative rainfall of UAF-RG station. The statistical analysis between
observed UAF-RG and signal based rainfall is analyzed. It is clear from the fig.4 that for L7,
L15, L18, L23, L22 and L29, as the distance increases 0.34, 3.39, 4.13, 6.21, 5.67 and 10.39
respectively from Reference UAF-RG, level of significance deceases. For validation data-set,
the coefficient of determination for L7, L15, L18, L23, L22 and L29, are 0.96, 0.79, 0.82, 0.78,
0.71 and 0.67 respectively. All the above results proved that rainfall is stochastic and erratic
pattern variable, as the distance increases from reference UAF-RG due to spatial variation,
rainfall depth increases or decreases. Similarly for all selected 35 links, as the distance increases




from the reference point UAF-RG, the spatial variability fluctuates due to spatial variation in
rainfall intensity.

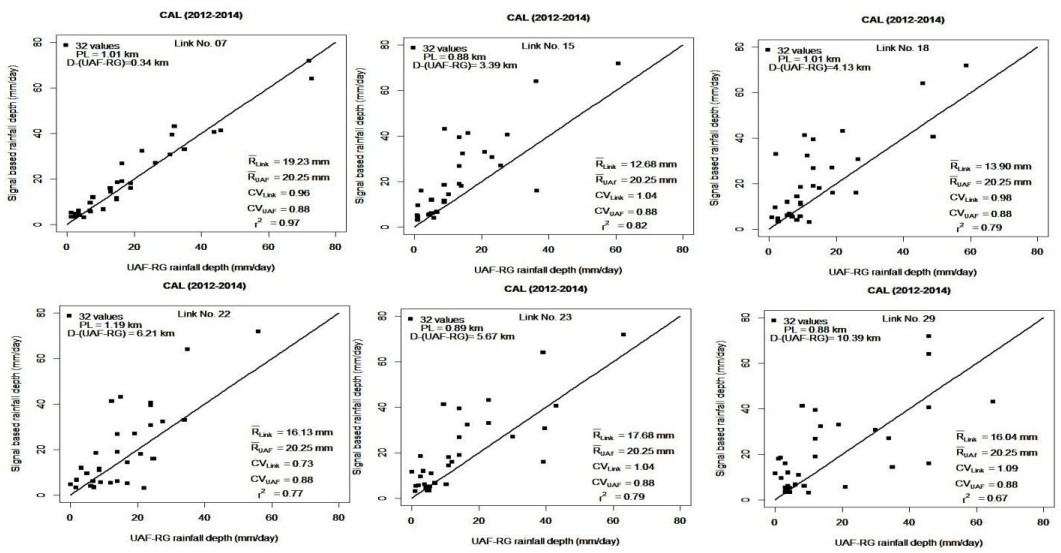


**Figure 4**. Scatter density plots of calibration data-set of daily (24hr) cumulative rainfall depths

of signal data of 33 no of days against daily cumulative rainfall depths of UAF rain gauge.

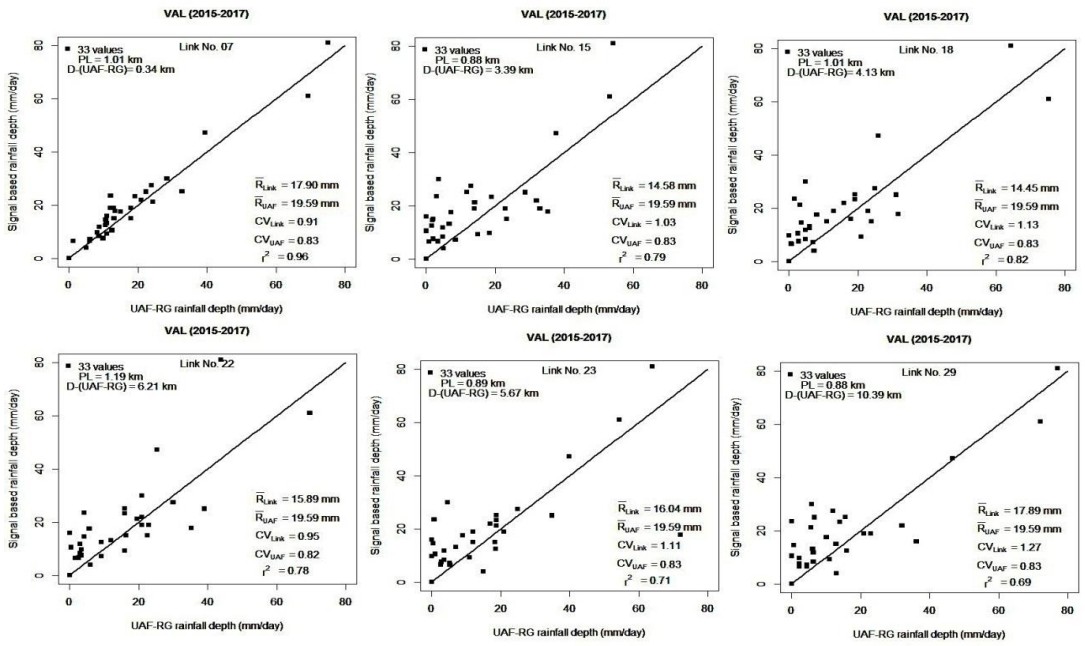




**Figure 5.** Scatter density plots of validation data-set of daily (24hr) cumulative rainfall depths of signal data of 33 no of days against daily cumulative rainfall depths of UAF rain gauge.

**4.3.2. Spatial percentage error analysis**

As it is discussed that there are mostly two sources available for rainfall estimation in Pakistan, which is rain gauge and satellite data. There are limited number of rain gauge networks operating in Indus basin irrigation system (IBIS), which is the largest irrigation system in the world, similarly rainfall estimated by satellite is also of low spatio-temporal resolution, so it is declared as data limited basin (Cheema 2012). Even the instruments which are installed on the existing meteorological stations in Pakistan are outdated and of low spatio-temporal resolution, so these low resolution data is used to estimated rainfall in basins and catchments, which is not the true presentation of the reality because rainfall is stochastic variable and its varies within radius of 1-2 km. Because of the factors discussed above, it is needs of the time that system should be established that provide high spatio-temporal resolution data, which is used for water resources management. Keeping in view of all these factors, rainfall is estimated by using signal based approach and by considering UAF-RG as reference point, percentage spatial error analysis is performed in the study area.

Figure 06 left (top and bottom) plots present percentage spatial variation of different links against no of rainy days from reference point UAR-RG. For calibration and validation data-set, the percentage spatial error for links no L2, L5, L7, L21, L25, and L28 varies between 20%-80% for different no of rainy days. It is clear from Fig. 06 left (top and bottom) plots that the percentage Error associated with spatial variation of rainfall from reference point (UAF-RG) i.e. the L2, L5 and L7 are close to reference UAF-RG, so there is small spatial error exist between these links, but the L21, L25 and L28 are far away from the UAF-RG reference point, so there is more spatial error exist. It is clear as the distance increase from the reference point point (UAF-RG) percentage error varies due to spatial variation of rainfall. Similarly fig.6 right (top and bottom) plots presents overall average percentage spatial error of all the selected links against the distance from reference UAF-RG. It is clear from the fig.6 right (top and bottom) plots that as the distance from the reference UAF-RG increases, for calibration and validation data-set, overall percentage average spatial error of all the selected links from reference UAF-RG varies between 10%-50% and 10%-60% respectively, which is logical and makes sense.





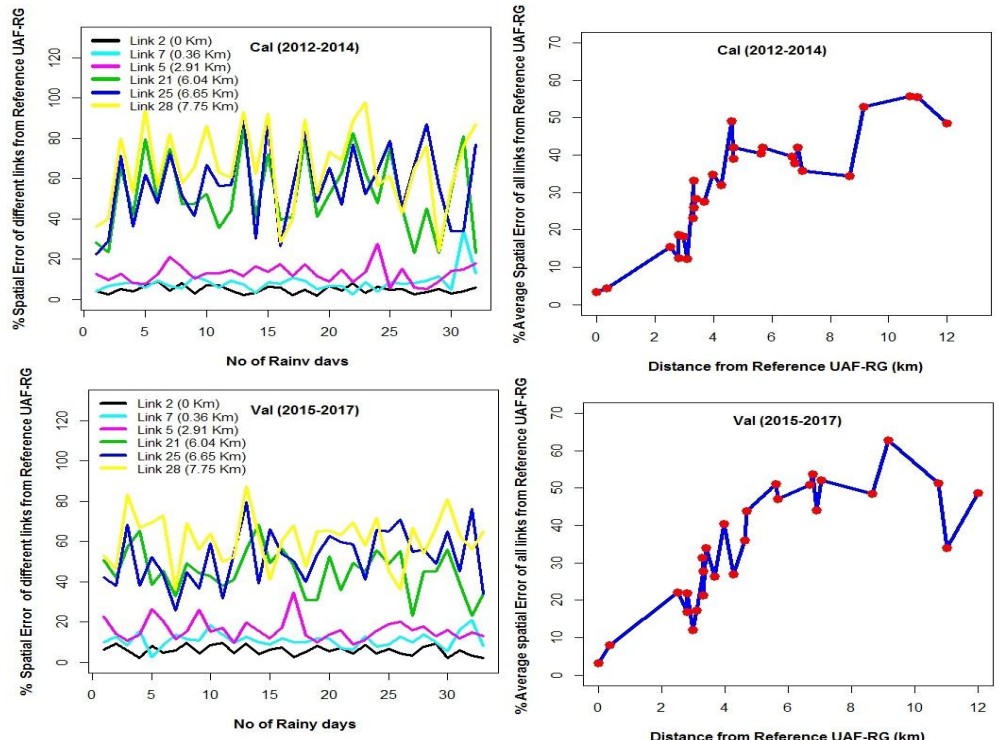

339

**Figure 6.** Left (top and bottom) plots present calibration and validation of percentage spatial error analysis of different links against no of rainy days. Right (top and bottom) plots present calibration and validation of percentage average spatial error of all links from reference UAF-RG.

**4.3.3. Rainfall mapping**
The rainfall maps are prepared in GIS by using IDW technique for rainfall event of 10
March 2014 and 23 July 2016. Figure 07 explains how rainfall varies within area of 225 $Km^2$.
The rainfall varies between 2 to 34 mm and 12 to 59 mm for 10 March 2014 and 23 July 2016
respectively and whereas rainfall recorded by UAF-RG on 10 March 2014 and 23 July May 2016
is 19mm and 40mm respectively and similarly rainfall recorded by WASA-RG on 10 March
2014 and 23 July May 2016 is 30mm and 26.3mm respectively. So one point based value is not a
presentation of whole study area because rainfall varies even within a distance of 1-2 km.



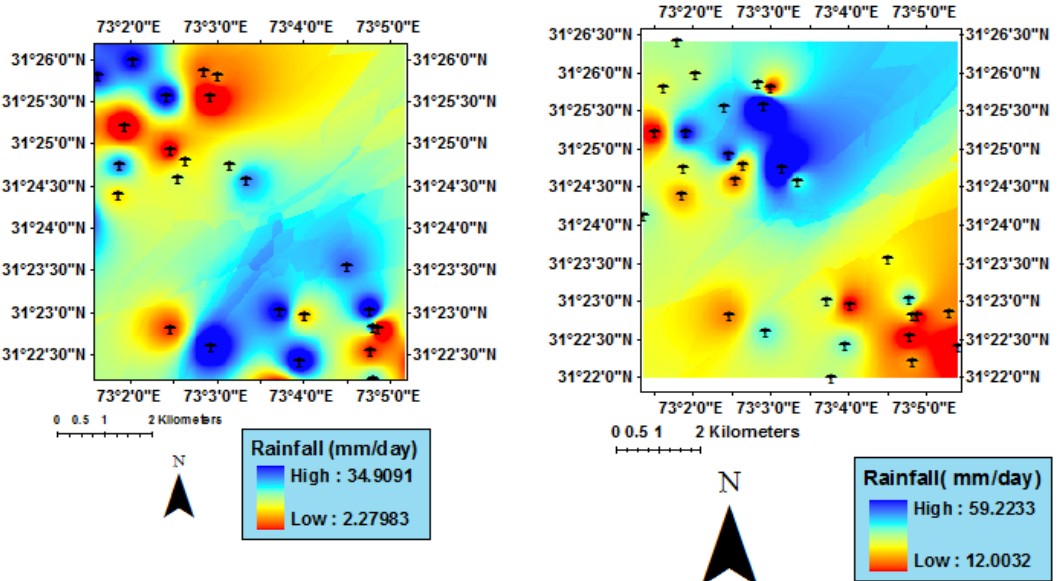


**Figure 7.** Signal based daily cumulative rainfall depths for all links on 10 March 2014 (left panel). Signal based cumulative rainfall depths for all links on 23 July 2016 (right plot). (Black tower represent different links location)

**5. Conclusion and recommendation**

The output of this paper shows that estimation of high resolution rainfall from the signal data of cellular communication network is the first step towards developing local climatic zones, which can contribute significantly in determining the components of water balance. The measurement of rainfall from the signal data of cellular communication network is a novel approach to obtain very high spatio-temporal resolution rainfall. The reason is that other resources such as satellite and rain gauges data cannot provide rainfall data for such a high 15 min resolution. There are total 97 rain gauges installed by PMD which are insufficient to capture high spatial and temporal resolution rainfall. Similarly satellite data which include data from TRMM having spatial resolution $0.25^{\circ}$ and minimum 3 hour temporal resolution, similarly GPM which is recently introduce having spatial resolution $0.1^{\circ}$ and 30 min temporal is not sufficient to presents the real situation. Therefore rainfall estimation by using signal processing is the needs



of hour and best suited in under developing country like Pakistan because readily existing setup
is used and there is no need of any extra cost of infrastructure or any setup required.
Commercial microwave links data of six years are used for rainfall estimation which
includes years 2012- 2014 for calibration and 2015-2017 for validation with UAF-RG, AR-RG
and WASA-RG respectively. This study proved that using microwave links for rainfall
estimation is very beneficial especially for both rural and urban areas. The algorithm developed
is highly low-cost and is a first step for the rainfall estimation from the signal data over the entire
area of Pakistan. The spatial error analysis also proved that rainfall is a stochastic variable. This
new technology has a great potential for calibration and validation for weather radar, assimilation
of different type of weather predication model or ground trusting of rainfall measured by using
satellite images.
This novel approach of measuring rainfall using the cellular communication network is the
Information and Communication Technology (ICT) revolution, which will definitely enhance the
role of ICT in agriculture and surface and groundwater resources on sustainable basis.
**Acknowledgements**
The authors would like to thanks Telenor Pakistan for providing the signal data and Dr Aart
Overeem, Dr Lenijine Hidde, Research Officers in KNMI Netherlands for providing the
technical support for processing of signal data. We are thankful to funding agency USAID

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
