# Peer review of "Manuscript under review for journal Hydrol. Earth Syst. Sci."

_Hydrology and Earth System Sciences, 2017_

## Referee Comment (RC1) · Anonymous Referee #1 · 24 Feb 2018

This paper is strongly based on Overeem, Aart, Hidde Leijnse, and Remko Uijlenhoet. "Retrieval algorithm for rainfall mapping from microwave links in a cellular communication network." Atmospheric Measurement Techniques 9.5 (2016): 2425-2444 (in the sequel, denoted by OLU). As I understood it, the authors applied the open code of OLU on measurements collected in a different geographic area, i.e. in Pakistan and analyzed the resulted performance of daily rainfall measurements. However, the methodology described in section 3 is far from being clear, and, in particular, deviations from OLU, if exist, are neither transparent not explained and justified. As such, the quality

and the significant of the results cannot be evaluated. Therefore, major revision of the manuscript is required to be able to properly review it for potential publication. In particular, a major revision is necessary with respect to the following issues: 1. Clarifying the claimed contribution. The claim contribution can be in empirical results for rainfall in Pakistan, or in presenting improvements to the algorithm of OLU, or in specifying challenges in applying it to an area different from Holland, where its PoC has been demonstrated. The authors must focus on their claim contributing so the paper can be evaluated accordingly. 2. Detailed and accurate description of the methodology. The methodology must be detailed to a level where it can be used by other. Unfortunately, in this paper the details are not even sufficient to evaluate the validity of the results. Just few examples (out of many!): The code developed by OLU required setting of many parameters. Which parameters have used in this paper? How are they set and why? In particular, which values of b and d were used in (3)? What was calibrated in section 3.2 and how? What is a "corrected maximum and minimum received power"? How are the rain maps validated? etc. 3. Major improvement of the writing and the presentation of the paper. Few examples: The abstract is too long; many terms were not defined; many details are missing, and more. In addition, there are numerous minor points that must be corrected. I will not specify them since the required major writing hopefully lead to fixing them.

---

## Short Comment (SC1) · 7 Mar 2018

Dear authors,

it was interesting to read about your research in Pakistan. It is good to see a further country where this new and promising technique is applied.

I just have two small comments:

1. The term "real time rainfall estimation" in the title is a bit misleading since you do

your analysis with a historic data set. Getting data in real-time is however possible (Chwala et. al 2016) and might increase the potential for applications in the future.

2. Most newer publications use the abbreviation "CML" (commercial microwave link) to refer to the microwave links, e.g. from cell phone networks, which we typically use to derive rainfall information.

Best regards, Christian Chwala

Chwala, C., Keis, F., and Kunstmann, H.: Real-time data acquisition of commercial microwave link networks for hydrometeorological applications, Atmos. Meas. Tech., 9, 991-999, https://doi.org/10.5194/amt-9-991-2016, 2016.

───────────────────────

---

## Short Comment (SC2) · 8 Mar 2018

Dear Authors, as the previous comments suggested, using the CML technology for rain estimation, and testing it especially in new regions is interesting and should be encouraged.

However, I think that some details of the estimation procedure should be better presented. Specifically: what values of the different parameters and/or coefficients you used exactly in the rain retrieval process, and how you calibrated them? Indeed,

Overeem et. al. (2016) algorithm is a well established one, but, it relies on parameters and coefficients (such as the power-law coefficients, the wet-antenna attenuation, etc.) which may have some variations from region to region (e.g., see (1))

In addition, in the manuscript conclusion, you express that: "The spatial error analysis also proved that rainfall is a stochastic variable". If you consider the rainfall to be a stochastic variable, then it might be interesting to consider discussing or even comparing your current work with recent models of rain estimation from CMLs which take advantage of the statistical properties of the rain-rate (e.g., (2))?

With Kind Regards,

Jonathan Ostrometzky.

1. Ostrometzky, J., Raich, R., Eshel, A., & Messer, H. (2016, March). Calibration of the attenuation-rain rate power-law parameters using measurements from commercial microwave networks. In Acoustics, Speech and Signal Processing (ICASSP), 2016 IEEE International Conference on (pp. 3736-3740). IEEE.

2. Ostrometzky, J., & Messer, H. (2018). Dynamic Determination of the Baseline Level in Microwave Links for Rain Monitoring From Minimum Attenuation Values. IEEE Journal of Selected Topics in Applied Earth Observations and Remote Sensing, 11(1), 24-33.

---

## Short Comment (SC3) · 21 Mar 2018

Dear Muhammad Sohail Afzal,

thank you for your response. Your future plans for setting up the real-time system, including the connection to agriculture, sounds very promising. But I guess there are still technical challenges to solve which are not part of the current manuscript. Hence, I still think that the term "real time" in the title is misleading. The details about your proposed system that you provided in your response, seem to give a great outlook,

though. It will certainly be very interesting to hear from your real-time system in the future.

Best regards, Christian Chwala

---

## Author Comment (AC1) · 21 Mar 2018

To the editorial board of HESS

Re:HESS-2017-740 "**Real time rainfall estimation using microwave signals of cellular communication networks: a case study of Faisalabad, Pakistan**"

Dear Reviewer

To begin with, I would like to thank you for considering our paper for your valuable comments. Below is response to your comments.

Best regards,
Muhammad Sohail Afzal

Comments of the reviewer/Reply

**Anonymous Referee #1**

**Comment 1**:
Clarifying the claimed contribution. The claim contribution can be in empirical results for rainfall in Pakistan, or in presenting improvements to the algorithm of OLU, or in specifying challenges in applying it to an area different from Holland, where it'sPoC has been demonstrated. The authors must focus on their claim contributing so the paper can be evaluated accordingly

*Reply:*
*The focus of this research is to adopt rainfall link based approachin under developed country like Pakistan, where other rain gauges or satellite data are limited or very expensive and not affordable. By testing the RAINLINK code developed in R language, we claimfollowing list of contributions, which are concluded below:*

1. *As Pakistan is under developed country,only 97 rain gauge stationsare operated by PMD (Pakistan Meteorological Department) Source (PMD website), which are very limited, even the rain gauges which are available are outdated and installed since 1970 and their validity has reduced with time.More than 70 % rain gauges are manual and remainingautomatic with systemic and personal errors. Similarly satellite data is also available butare very expensive with high spatio-temporalresolution, therefore, most of the researchers and scientists depend only on the PMD rainfall data. Some institutions installed their own rain gauge in their regional offices without considering the TOR of the installation and subsequently rainfall data manifest biasness.This link based approach introduced in Pakistan is unique and will help in real time monitoring of different studies. The main focus and*

*contribution of this conducted research is to adopt and promote this new method of rainfall estimation in Pakistan.*

2. *After adoption of this new method of rainfall estimation on high spatial and temporal resolution, this method can be further used for different studies including real time irrigation scheduling and to develop a system, as it is working in California "California Irrigation Management Information system(CIMIS)",flood monitoring and forecasting, climate changeadoption and sustainable groundwater management.*

3. *The operating frequency of the signal used is 38 GHz and signal is vertically polarized having 15 min temporal resolution, which is the first initial point to move further investigation because the same operating frequency is used in Holland with vertical polarization. The detaileddescription of parameters especially a and b chosenis given in reply of comment 2.*

4. *In Faisalabad, wind speed is insignificant and attenuation in the signal is purely due to rainfall intensity therefore values ofaand b have been calibrated and validated (see comment 2). All the observed rainfall in the three observation points matches with the signal based rainfall at the corresponding location of observed rain gauges station. Also the statistical measure mean, SD, CV,and $R^2$ represent good agreement between observed and signal based rainfall.*

5. *To study the spatial rainfall analysis, RAINLINK code is run which confirms that rainfall is a stochastic variable and it shows variations within 1-2 km range.*

6. *In future studies, calculation of a and b for different geographical location of Pakistan will be tested because rainfall varies between 250-1500 mm in different area of Pakistan.Of course, the rainfall intensity is affected by wind speed, attitude and temperature. In next paper, we are working to derive the values of a and b at different attitudes in Pakistan. Again, we believe that in Faisalabad, the climate is very stable and values of a and b have been fitted/calibrated based on the vertical polarization and 38 GHz frequency and shows good agreement between observed and predicted values.*

7. *Based on the sensitivity of the algorithm, it is concluded that smaller path length should be selected for link based approach for best results of rainfall estimation along with high frequency.*

**Comment 2**:

Detailed and accurate description of the methodology. The methodology must be detailed to a level where it can be used by other. Unfortunately, in this paper the details are not even sufficient to evaluate the validity of the results. Just few examples (out of many!): The code developed by OLU required setting of many parameters. Which parameters have used in this paper? How are they set and why?In particular, which values of b and d were used in (3)? What was calibrated in section 3.2 and how? What is a "corrected maximum and minimum received power"? How is the rain maps validated? Etc

- Detailed and accurate description of the methodology. The methodology must be detailed to a level where it can be used by other

*Reply:*

*Detail stepwise algorithm methodology regarding different rainfall retrieval steps included preprocessing; dry-wet classification, references signal level etc will in cooperated in final revised manuscript. Initially we provided the detailed methodology firs each step of rainfall processing but in initial submission according to the editor directions, we change the methodology into abstracted form. Now, revised manuscript has been incorporated with detailed methodology.*

- The code developed by OLU required setting of many parameters. Which parameters have used in this paper? How are they set and why? In particular, which values of b and d were used in

*Reply:*

*There are different parameters used in this study. Some parameters are fixed and recommended by Overeem et al 2016 to test in different geographical areas of the world. For exampleattenuation due to wet antenna $A_a=2.3$ and coefficient $\alpha=0.33$. Among all the parameters, values of c=a and d=b for local climatic conditions are also very important.*

*The snapshot of small section of input file used to process through RAINLINK codeis given below in Figure1. Fromthis Figure, it is clear that the frequency of link that has been used in this study is 38GHz and the signal is vertically polarized. The same operating frequency link has been used in Holland. The Figure 2 hasbeen used as reference to fit/calibrate the values of **c=3.69 and d=1.04.**These values have been selected after a number of simulations until the observed and simulated values match each other. The output file during the simulation of rainfall algorithm is shown in Figure 3 which same values of c and d for each time step (15 min interval)and these values have been finalized after a number of simulation.*

| Frequency | YearMonthDaytime | RxLevelMin | RxLevelMax | path -length | x-start | y-start | x-end | y-end | Link ID | Link ID |
|---|---|---|---|---|---|---|---|---|---|---|
| 38 | 201606220015 | -41.8 | -43.3 | 0.88 | 31.4147 | 73.0276 | 31.41 | 73.035 | 1 | FFD316-FFD309 |
| 38 | 201606220030 | -41.8 | -42.7 | 0.88 | 31.4147 | 73.0276 | 31.41 | 73.035 | 1 | FFD316-FFD309 |
| 38 | 201606220045 | -41.8 | -42.7 | 0.88 | 31.4147 | 73.0276 | 31.41 | 73.035 | 1 | FFD316-FFD309 |
| 38 | 201606220100 | -41.8 | -42.5 | 0.88 | 31.4147 | 73.0276 | 31.41 | 73.035 | 1 | FFD316-FFD309 |
| 38 | 201606220115 | -41.8 | -42.5 | 0.88 | 31.4147 | 73.0276 | 31.41 | 73.035 | 1 | FFD316-FFD309 |
| 38 | 201606220130 | -41.8 | -42.5 | 0.88 | 31.4147 | 73.0276 | 31.41 | 73.035 | 1 | FFD316-FFD309 |
| 38 | 201606220145 | -41.8 | -42.5 | 0.88 | 31.4147 | 73.0276 | 31.41 | 73.035 | 1 | FFD316-FFD309 |
| 38 | 201606220200 | -41.6 | -42.5 | 0.88 | 31.4147 | 73.0276 | 31.41 | 73.035 | 1 | FFD316-FFD309 |
| 38 | 201606220215 | -41.6 | -42.5 | 0.88 | 31.4147 | 73.0276 | 31.41 | 73.035 | 1 | FFD316-FFD309 |
| 38 | 201606220230 | -41.8 | -42.5 | 0.88 | 31.4147 | 73.0276 | 31.41 | 73.035 | 1 | FFD316-FFD309 |
| 38 | 201606220245 | -41.6 | -42.3 | 0.88 | 31.4147 | 73.0276 | 31.41 | 73.035 | 1 | FFD316-FFD309 |
| 38 | 201606220300 | -41.6 | -42.3 | 0.88 | 31.4147 | 73.0276 | 31.41 | 73.035 | 1 | FFD316-FFD309 |
| 38 | 201606220315 | -41.6 | -42.5 | 0.88 | 31.4147 | 73.0276 | 31.41 | 73.035 | 1 | FFD316-FFD309 |
| 38 | 201606220330 | -41.6 | -42.5 | 0.88 | 31.4147 | 73.0276 | 31.41 | 73.035 | 1 | FFD316-FFD309 |
| 38 | 201606220345 | -41.6 | -42.5 | 0.88 | 31.4147 | 73.0276 | 31.41 | 73.035 | 1 | FFD316-FFD309 |
| 38 | 201606220400 | -41.6 | -42.5 | 0.88 | 31.4147 | 73.0276 | 31.41 | 73.035 | 1 | FFD316-FFD309 |
| 38 | 201606220415 | -41.6 | -42.5 | 0.88 | 31.4147 | 73.0276 | 31.41 | 73.035 | 1 | FFD316-FFD309 |

Figure 1.

[Figure]

**Figure 2.** Values of coefficients in the relationship to convert specific attenuation to rainfall intensity for frequencies ranging from 6 to 50 GHz. The grey-shaded area denotes the 37.0–40.0 GHz range. Note the logarithmic vertical scale in the left figure. Here values have been computed from one data set of measured drop size distributions (p. 65 in Leijnse (2007c); solid lines). The values recommended by the International Telecommunication Union (ITU, 2005), meant for computing specific attenuation for given rain rates and for worldwide application, are also plotted (dashed and dotted lines).

- What was calibrated in section 3.2 and how?

*Reply:*

*In section 3.2, two different things were calibrated and validated, firstly input parameters and secondly signal based rainfall estimated from RAINLINK code was calibrated and validated with local rain gauges data available.*

*Calibration of Input parameters. After selection of all input parameters which include c, d, $A_a$ and α etc as described above (reply of comment 2), all the parameters have been used initially to test the sensitivity of algorithm and the RAINLINK code has been run*

*forcalibrationtime period of32 days(including rainy and non rainy days) from year 2012-2014. The RAINLINK code has been run for different number of days and value of c and d have been checked in the output file of RAINLINK code, which are in agreement with the local condition values as shown in Figure 2. This represents that values of c and d have been calibrated as rainfall calculated from RAINLINK code and observed rainfall represents very good agreement. After final calibration of RAINLINK code in term of (signal based rainfall with local rain gauges data) then the all the parameter were fixed and run the RAINLINK code for 33 days (including rainy and non rainy days) for validation purpose for year 2015-2017. The rainfall estimated from RAINLINK code for validation period also represents good agreement of observed rainfall with the signal based calculated rainfall. This shows that fitted parameters in the calibration process have been validated for selected site in Faisalabad.*

- What is a "corrected maximum and minimum received power

***Reply:***

*The Figure 3 is used to response this comment. The corrected maximum and minimum received powers have been calculated for maximum and minimum power which is provided as input data and also shown in Figure 1(**received maximum power (Rmax) and received minimum power (Rmin)**). A detail stepwise procedure has been used to first calculate reference signal power and then compared the received maximum and minimum power with reference signal to calculatecorrected maximum and minimum power which is described below. The Figure 3 shows the final output file of all the parameter including maximumreceivedpower (Rmax), received minimum power (Rmin), reference signal power (ref_level)and corrected maximum power (Rmax_final) and corrected minimumpower (Rmin_final). The graphical representation of this corrected power has been presented in Figure 2 in the manuscript for just 3 links (**L1, L7 and L19**).*

[Figure]

Figure 3

*Detail methodology which has been adopted is listed below to calculate corrected*
*maximum and minimum power and then rainfall*

**Power law**

*Microwave links are the main source of communication between the*
*telecommunication towers.In these links electromagnetic signal move from one*
*antenna of one tower(transmitter) to the other antenna of the other tower( receivers)*
*and during rainfall these sending signal from one tower to other tower attenuate due to*
*rainfall intensity [Upton et al., 2005]. This attenuation of the signal due to rainfall*
*intensity can be measured by the state of the art using power law [Atlas and Ulbrich ,*
*1977] which is the relationship between rainfall and specific attenuation which is*
*given as below :*

$$R = ck^d \qquad (1)$$

*In the above equation k is the specific attenuation, R is the intensity of rainfall(mmh⁻*
*¹), c is the coefficient and d is the exponent and the valves of these coefficient and*
*exponent depend upon frequency and polarization mainly and these parameters have*
*been described above (**Reply of comment 2**). The final form of rainfall intensity is*
*given below and detailed explanation will be presented in final revised manuscript*

$$< R >= c \ [\frac{Fref(L)-F(L)}{L}] \qquad (2)$$

***Pre- processing:***

*This is the first step in which input file is corrected on the basis certain checks, which is readily discussed in paper.***In pre-processing*** *file free from errors including unique time interval having multiple records, missing link data for the selected unique link,and variation in frequency, link coordinates, path length for a unique link is prepared.*

***Dry-Wet classification:***

*The second step in rainfall estimation for signal is to distinguish between the dry and wet signal.The stepwise procedure how to differentiate between the dry and wet spells is given below (Overeem et al., 2016):*

1. *In first step select a link having 24 hours period.*
2. *All the links should be less than 15 km from the selected link.*
3. *Continuous to select the link within range of 15 km of selected link having all data available of present and pervious day. There should be at least 3 links within range of 15 km otherwise neglect the selected link and go to first step to select another link.*
4. *Now calculate the specific attenuation $\Delta F_L = \frac{Fmin - max(Fmin}{L}$and calculate the median values of all calculated values $\Delta F$ and $\Delta F_L$ of all the selected links.*
5. *Now there is a criteria that if the median value of $\Delta F_L$ is less than -0.7 dBkm$^{-1}$ and median value of $\Delta F$ is less than -1.4 dB then the signal is classified as wet signal .*
6. *All time intervals that have not been classified as wet are classified as dry for them link selected in step 1.*
7. *Do the same procedure for the other entire link having time interval of 24 hours period.*

***Reference Signal Level and corrected received power***

*The path average rainfall which is calculated by the differentiation between the maximum and minimum received power and the reference signal power which is selected on the day weather condition. The reference signal is calculated for all the selected links and for all 15 min resolution time interval by taking the median of the time interval for present and pervious 24 hours time period which is classified as dry weather condition. After this now rainfall is not estimated from signal if dry days having 15 min resolutions are less than 1 that is 2.5 h in 24h. Corrected signals $F^C_{min}$ are then applied on which next analysis is performed:*

$$F^C_{min} = Fmin \quad if\ wet\ AND\ Fmin < Fref \quad Eq.(3)$$
$F_{ref}$ *if dry OR* $F_{min} \geq F_{ref}$

*Subsequently, the corrected maximum received power is calculated as follows:*
*Similarly corrected power received is estimated as below formula*

$$F^C_{max=} \quad Fmax \quad if \ F^C_{min} < Fref \ AND \ Fmax < Fref \quad Eq.(4)$$
$$Fref \quad if \ F^C_{min} \ = Fref \ OR \ Fmax \ \geq \quad Fref$$

**Determination of path average rainfall intensity**

*In this step rainfall has been calculated from the maximum corrected received power and minimum corrected received power with 15 min resolution. In this step the attenuation due to wet antenna and temporal variation also taken into account. The rainfall intensity due to attenuation of the signal can be calculated by using following relation given below*

$$Amin = Fref \ - F^C_{max} (5)$$

$$Amax = Fref \ - F^C_{min} (6)$$

$$kmax = \frac{Amax - Aa}{L} H(A_{max}\text{-}A_a) \qquad (7)$$

$$kmin = \frac{Amin - Aa}{L} H(Amin - Aa)(8)$$

$$< R > = \alpha a k^b_{max} + (1 - \alpha) \ a \ k^b_{min} \qquad (9)$$

*Where $k_{min}$ and $k_{max}$ are the maximum and minimum attenuation in decibel per kilometer, Aa is attenuation due to wet antenna, α is a coefficient that is calculated by the contribution of the minimum and maximum attenuation having 15 min resolution and H is the Heaviside function (if the argument of H is smaller than zero, H = 0, else H = 1).*

*Note: all the above explained steps are scripted in RAINLINK code which is written in R language and modified according to our input data. Detail explanation of all above steps has been included in the revised final manuscript.*

- How is the rain maps validated?

***Reply:***
*Rainfall maps has been developed in ArcMap by interpolation techniquei.e IDW and for validation purpose, three raingauges(UAF-RG, WASA-RG, WMRC-RG and AR-RG) have been located within study area and used for validation of obserbed and simulated rainfall. The results have been incorporated in the final revised manuscript.*

*The detail procedure which has beenadopted to validate the rainfall maps given below*

1. *Rainfall map has been prepared in Arcmap by IDW interpolation technique*
2. *Shape file of three selected rain gauges has been prepared*
3. *The shape file of selected rain gauges have been added on the rainfall map in arc map*
4. *The shape file of selected rain gauges exactly overlaps the rainfall map*
5. *Then with the help of cursor, predicated rainfall values against each pixel where the rain gauges exactly located are noted.*
6. *Statistical analyses in term of mean, RMSE, coefficient of variance and coefficient of determination $R^2$ has been performed between the observed rainfall values from rain gauges and pixel information from rainfall maps against each overlap rain gauge has been recorded.*
7. *By adopting this procedure, revised manuscript has been set accordingly.*

**Comment 3:**
Major improvement of the writing and the presentation of the paper. Few examples: The abstract is too long; many terms were not defined; many details are missing, and more. In addition, there are numerous minor points that must be corrected

*Reply:*

*Yes, we have incorporated your comments to further improve the abstract and given detailed information about the missing parameters. Minor corrections have also been incorporated in the revised final manuscript.*

.

---

## Author Comment (AC2) · 21 Mar 2018

To the editorial board of HESS

Re: HESS-2017-740 "**Real time rainfall estimation using microwave signals of cellular communication networks: a case study of Faisalabad, Pakistan**"

Dear Christian Chwala

To begin with, I would like to thank you for your consideration of our paper. The comments you have given, made me to reconsider the paper on basic aspects.

Best regards,
Muhammad Sohail Afzal

Comments of the reviewer/Reply

**Comment 1:**
The term "real time rainfall estimation" in the title is a bit misleading since you do your analysis with a historic data set. Getting data in real-time is however possible (Chwala et. al 2016) and might increase the potential for applications in the future.

*Reply*
*Yes we agree that we did analysisfor that historic data for period 2012-2017. No we have also calculated rainfall for 2017(May onward) and 2018 on ward. Of course we will include rainfall estimation for current date, and then it will become real time. Later on, we are working to develop framework for real time rainfall estimation with collaboration of Telenor Telecommunication Company. The point is that the farmers will be registered with Telenor Moblie agriculture service and farmers will receive the message of rainfall at the specific location.Further we are working on developing real time irrigation management information system (IMIS) by developing mobile application with government of Punjab,Pakistan. This rainfall information will be integrated in the mobile app for real time estimation of rainfall which will help farmer to perform real time irrigation scheduling.*

**Comment 2:**
Most newer publications use the abbreviation "CML" (commercial microwave link) to refer to the microwave links, e.g. from cell phone networks, which we typically use toderive rainfall information.

*Reply*
*Yes we agree that CML (commercial microwave link) is latest terminology used to derive rainfall information. We will incorporate this word CML in the updated version of the manuscript.*

---

## Author Comment (AC3) · 21 Mar 2018

To the editorial board of HESS

Re: HESS-2017-740 "**Real time rainfall estimation using microwave signals of cellular communication networks: a case study of Faisalabad, Pakistan**"

Dear Jonathan Ostrometzky

To begin with, I would like to thank you for your consideration of our paper. The comments you have given, made me to reconsider the paper on basic aspects.

Best regards,
Muhammad Sohail Afzal

Comments of the reviewer/Reply

**Comment 1:**
I think that some details of the estimation procedure should be better presented.
Specifically: what values of the different parameters and/or coefficients you used exactly in the rain retrieval process, and how you calibrated them? Indeed, Overeem et. al. (2016) algorithm is a well established one, but, it relies on parameters and coefficients (such as the power-law coefficients, the wet-antenna attenuation, etc.) which may have some variations from region to region (e.g., see (1))

*Reply*
*See reply of RC1 for detailed information.*

**Comment 2:**
In addition, in the manuscript conclusion, you express that: "The spatial error analysis also proved that rainfall is a stochastic variable". If you consider the rainfall to be a
stochastic variable, then it might be interesting to consider discussing or even comparing
your current work with recent models of rain estimation from CMLs which take advantage of the statistical properties of the rain-rate (e.g., (2))?

Reply
*Yes, your comment is very important to compare the signal based rainfall with other statistical rainfall models for verifying the stochasticity. We will discuss in the manuscript the stochasticity nature of the rainfall. Basically rainfall follows the exponential distribution and shows erratic pattern having probabilistic distribution and hence rainfall is stochastic.*

---

## Referee Comment (RC2) · Anonymous Referee #2 · 4 Apr 2018

First of all I would like to congratulate the authors to have initiated this work on Commercial Microwave Links (CMLs) for rainfall measurement in Pakistan and obtained data from a telecom compagny. The hardest is done ! More work is now needed to provide a quantitative evaluation of the CML based estimation and explain in more details the data processing.

The main points to be worked on in order to bring this work to a publishable paper are :

METHOD : - the data processing needs to be detailed. The authors used an existing and open source code provided by Overeem/Leijnse/Uijlenhoet, however the proposed algorithm needs to be tuned and some parameters set (for instance the exponent and prefactor of the attenuation/rainfall relationship in eq 1). The authors have to explain how they did this, what choices were made and which uncertainties were analyzed. Given that the evaluation is done on a daily time step basis, while the data is gathered at 15 minutes, are the parameters set up at 15 ' or daily time step ? -They also need to provide more details on the way they calculate the baseline or reference attenuation. And what they call ' corrected' attenuations 'free of errors' (p6 l 150). -Also they mention the use in the processing of several neighboring links, with a range of operating frequencies ; they should explain how their processing is adapted to this variation in CML characteristics within the network.

TIME RESOLUTION -In the introduction and through the text the concept of high space-time resolution is put forward. However the evaluation is provided at the daily time step – mention of high time resolution should be suppressed. -'Real time ' should be suppressed from the title as the work is based on archived data. The RT prospect can of course be mentioned in the perspectives.

SPATIAL ANALYSIS Using the CML density to analyze the spatial structure of the rain field is an original idea. I encourage the authors to clarify and further develop the analysis presented in Fig 6.

ENGLISH TEXT : Once the content of the work has been improved special care should be taken in the writing. But let's work one step at a time.

Given the comments above, I encourage the authors to submit a substantially revised manuscript with a detailed description of their methodology and quantitative results.

---

## Author Comment (AC4) · 13 Apr 2018

To the editorial board of HESS

Re: HESS-2017-740 "**Real time rainfall estimation using microwave signals of cellular communication networks: a case study of Faisalabad, Pakistan**"

Dear Reviewer

To begin with, I would like to thank you for your consideration of our paper. The comments you have given, made me to reconsider the paper on basic aspects.

Best regards,
Muhammad Sohail Afzal

Comments of the reviewer/Reply

**Anonymous Referee #2**

**Comment 1:**
METHOD: - the data processing needs to be detailed. The authors used an existing and open source code provided by Overeem/ Leijnse/ Uijlenhoet, however the proposed algorithm needs to be tuned and some parameters set (for instance the exponent and pre factor of the attenuation/rainfall relationship in eq1). The authors have to explain how they did this, what choices were made and which uncertainties were analyzed. Given that the evaluation is done on a daily time step basis, while the data is gathered at 15 minutes, are the parameters set up at 15 ' or daily time step ? -They also need to provide more details on the way they calculate the baseline or reference attenuation. And what they call 'corrected' attenuations 'free of errors' (p6 l 150). - Also they mention the use in the processing of several neighboring links, with a range of operating frequencies; they should explain how their processing is adapted to this variation in CML characteristics within the network.

- The data processing needs to be detailed. The authors used an existing and open source code provided by Overeem/ Leijnse/ Uijlenhoet, however the proposed algorithm needs to be tuned and some parameters set (for instance the exponent and pre factor of the attenuation/rainfall relationship in eq1). The authors have to explain how they did this, what choices were made and which uncertainties were analyzed.

*Reply/ See response of RC1, Comment-2 for detailed information about input parameters that have been used in this study. Further calibration of alpha and Aa (wet antenna attenuation) values have also been incorporated in the final revised manuscript [Overeem (2013, 2016)].*

- Given that the evaluation is done on a daily time step basis, while the data is gathered at 15 minutes, are the parameters set up at 15 ' or daily time step ?

*Reply/ Yes all the parameters are set at 15 min time step to estimate rainfall at 15 min temporal resolution, which is converted into commutative rainfall representing one value at daily time step. Furthermore paper presents analysis on daily time step basis rainfall as the observed rainfall for the observed rain gauges in Faisalabad (UAF-RG, WASA-RG and AR-RG) is available on a daily time step.*

- They also need to provide more details on the way they calculate the baseline or reference attenuation. And what they call 'corrected' attenuations 'free of errors' (p6 l 150). -Also they mention the use in the processing of several neighboring links, with a range of operating frequencies; they should explain how their processing is adapted to this variation in CML characteristics within the network.

*Reply/ See the response to RC1, comment-2 for detail information and more detail with different results has been in cooperated in the final revised manuscript.*

**Comment 2:**

TIME RESOLUTION -In the introduction and through the text the concept of high space-time resolution is put forward. However the evaluation is provided at the daily time step – mention of high time resolution should be suppressed. -'Real time 'should be suppressed from the title as the work is based on archived data. The RT prospect can of course be mentioned in the perspectives.

*Reply/ The short comment (SC1) from the reviewer also mentioned the same reservation which is valid. Accordingly we have changed the title and mentioned in the text the prospective of the real time rainfall estimation.*

**Comment 3:**

SPATIAL ANALYSIS Using the CML density to analyze the spatial structure of the rain field is an original idea. I encourage the authors to clarify and further develop the analysis presented in Fig 6.

*Reply/ Spatial variation analysis of rainfall with respect to reference rain gauges has been performed to confirm stochasticity of rainfall using signal data of CML. The further in depth analysis at higher than 15 min temporal resolution would definitely make the rainfall structure and will be helpful for analyzing cause-effects relationship. As the signal data available with the telecommunication company is 15 minutes temporal resolution, therefore the presented analysis gives very good picture about the erratic pattern of the rainfall and leads towards further analysis related with "Rainfall estimation and predication using signal based rainfall in Punjab, Pakistan" and "Redefining rainfall distribution using signal based rainfall and modeling approach in Pakistan", which are forthcoming papers. Also long term trends of stochasticity nature of the rainfall would guide us to determine the band strip of maximum and minimum*

*rainfall, average rainfall at a site specific location, which will be value added information for irrigation and fertigation scheduling.*

**Comment 4:**
Once the content of the work has been improved special care should be taken in the writing. But let's work one step at a time.

*Reply/ Yes, we have incorporated your comments to further improve quality of paper and presentation of the paper with improved writing.*

---

## Author Comment (AC5) · 13 Apr 2018

To the editorial board of HESS

Re: HESS-2017-740 "**Real time rainfall estimation using microwave signals of cellular communication networks: a case study of Faisalabad, Pakistan**"

Dear Christian Chwala

To begin with, I would like to thank you for your consideration of our paper. The comments you have given, made me to reconsider the paper about the title.

Best regards,
Muhammad Sohail Afzal

**Short Comment (SC1)**

**Comment 1:**
Your future plans for setting up the real-time system, including the connection to agriculture, sounds very promising. But I guess there are still technical challenges to solve which are not parts of the current manuscript. Hence, I still think that the term "real time" in the title is misleading. The details about your proposed system that you provided in your response seem to give a great outlook, though. It will certainly be very interesting to hear from your real-time system in the future.

*Reply/* *Your reservation "Real time" in the title is valid. So we have changed the title and mentioned in the text the prospective of the real time rainfall estimation.*